# Chemopreventive Activity of Ellagitannins from *Acer pseudosieboldianum* (Pax) Komarov Leaves on Prostate Cancer Cells

**DOI:** 10.3390/plants12051047

**Published:** 2023-02-24

**Authors:** Se-Yeon Son, Jin-Hyeok Choi, Eun-Bin Kim, Jun Yin, Seo-Yeon Seonu, Si-Yeon Jin, Jae-Yoon Oh, Min-Won Lee

**Affiliations:** Laboratory of Pharmacognosy and Natural Product Derived Medicine, College of Pharmacy, Chung-Ang University, Seoul 06974, Republic of Korea

**Keywords:** *Acer pseudosieboldianum* (Pax) Komarov, ellagitannins, chemopreventive activity, prostate cancer, demethylation activity

## Abstract

Several studies have shown that compounds from *Acer pseudosieboldianum* (Pax) Komarov leaves (APL) display potent anti-oxidative, anti-inflammatory, and anti-proliferative activities. Prostate cancer (PCa) is the most common cancer among older men, and DNA methylation is associated with PCa progression. This study aimed to investigate the chemopreventive activities of the compounds which were isolated from APL on prostate cancer cells and elucidate the mechanisms of these compounds in relation to DNA methylation. One novel ellagitannin [komaniin (**14**)] and thirteen other known compounds, including glucose derivatives [ethyl-β-D-glucopyranose (**3**) and (4R)-p-menth-1-ene-7,8-diol 7-O-β-D-glucopyranoside (**4**)], one phenylpropanoid [junipetrioloside A (**5**)], three phenolic acid derivatives [ellagic acid-4-β-D-xylopyranoside (**1**), 4-O-galloyl-quinic acid (**2**), and gallic acid (**8**)], two flavonoids [quercetin (**11**) and kaempferol (**12**)], and five hydrolysable tannins [geraniin (**6**), punicafolin (**7**), granatin B (**9**), 1,2,3,4,6-penta-galloyl-β-D-glucopyranoside (**10**), and mallotusinic acid (**13**)] were isolated from APL. The hydrolyzable tannins (**6**, **7**, **9**, **10**, **13**, and **14**) showed potent anti-PCa proliferative and apoptosis-promoting activities. Among the compounds, the ellagitannins in the dehydrohexahydroxydiphenoyl (DHHDP) group (**6**, **9**, **13**, and **14**), the novel compound **14** showed the most potent inhibitory activity on DNA methyltransferase (DNMT1, 3a and 3b) and glutathione S-transferase P1 methyl removing and re-expression activities. Thus, our results suggested that the ellagitannins (**6**, **9**, **13**, and **14)** isolated from APL could be a promising treatment option for PCa.

## 1. Introduction

Prostate cancer (PCa), the cancer of the prostate gland, is the most common cancer in older men between the age of 54 to 75 years in Western countries. DNA methylation activity occurs in several cancer types, including PCa. DNA methylation is a process where methyl groups are added to the cytosine, creating a methylated cytosine, and DNA methylation can change DNA segment activity without changing the sequence. DNA methylation typically inhibits gene transcription, which is located in the gene promoter regions, and this process is called gene silencing. Due to the activity of gene silencing, a number of key processes occur, including X-chromosome inactivation, inhibition of transposable elements, aging, and carcinogenesis [1,2,3,4,5]. Oxidative stress, inflammation, cytokine release, and transcription factor nuclear factor kappa-light-chain-enhancer of activated B cells (NF-κB) can catalyze DNA methylation via the upregulation of DNA methyltransferases (DNMTs) in PCa [6,7,8,9,10,11,12,13,14].

DNMTs, including DNMT1, DNMT3a, and DNMT3b, are the family of enzymes that catalyze the transfer of methyl group to DNA [15,16,17]. DNMT1 is the most abundant DNMT and contributes to both de novo and maintenance methylation on tumor suppressor genes, such as GSTP1, etc. However, the activity on hemimethylated DNA of DNMT1 is more potent than de novo [18,19,20]. The DNMT3 enzymes, including DNMT3a and DNMT3b, are mainly responsible for active de novo methylation [21,22]. Although DNMT1 was considered to be highly expressed in cancer cells, the increase in expression of de novo methyltransferases, DNMT3, also have been proven to be involved with cancer cells [23,24,25]. There are many papers that proved oxidation, inflammation, IL-6, and NF-κB can increase DNMTs and induce DNA methylation [6,7,8,9,10,11,12,13,14]. Thus, inhibition of these factors may be the key to DNA demethylation activity.

GSTP1 belongs to the Glutathione S-transferases (GSTs) enzymes family that plays a key role in detoxification by conjugation of hydrophobic and electrolytic components. The GSTs are categorized into four classes, alpha, mu, pi and theta, and these enzymes affect several factors, such as signaling pathway modulating involved in cell proliferative, progression, tumor development and recurrence. Methylation of the GSTP pi gene (GSTP1) is associated with tumor development including neuroblastoma, hepatocellular carcinoma, and endometrial, breast and prostate cancers. Researchers reported that GSTP1 is hypermethylated in prostate cancer, while benign prostatic hyperplasia and normal prostate cells are hypomethylated [26,27,28,29]. The GSTP1 methylation of prostate cancer has been recommended as an epigenetic marker by many researchers.

Inflammation has been suggested to be a key factor in the development of several tumors, including PCa. It is extensively considered that the pro-inflammatory cytokines (such as tumor necrosis factor α, interleukin [IL]-6, and IL-1) and anti-inflammatory cytokines (such as epidermal growth factor and transforming growth factor beta) play a causative role in the carcinogenesis of PCa [30,31,32,33,34,35]. IL-6 plays a key role not only in advancing tumor progression but in the early stages of carcinogenesis. Moreover, Okamoto et al. reported that IL-6 could enhance both androgen-dependent and independent cell (LNCaP and PC-3) growth but not the growth of benign prostatic hyperplasia [36,37,38,39,40]. IL-6 also induces DNMTs activity, thus playing a regulatory role in DNA methylation in cancer [8,10]. Therefore, the specific inhibition of IL-6 could impede PCa growth.

NF-κB is a transcription factor that mediates various functions, such as regulating DNA transcription, cytokine production, and cell survival. It also reacts with free radicals, cytokines, UV irradiation, and antigens [41,42,43]. The dysregulation of NF-κB has been associated with inflammation, improper immune development, and cancer. Many publications on different types of human tumors, including PCa, have reported that the dysregulation of NF-κB contributes to cell proliferation and survival. Normal cells die when nutrients or energy are depleted through the regulation of NF-κB, but this does not occur in cancer cells owing to its anti-apoptosis activity [44,45,46,47,48]. Moreover, NF-κB is regulated by DNA methylation [12]. Thus, a compound that inhibits NF-κB was proposed to be an effective cancer therapy and prevention agent.

Apoptosis is a form of programmed cell death, that is not similar to necrosis in which cells die due to acute injury. There are two pathways in the apoptosis activation mechanism, intrinsic and extrinsic pathways, leading to the activation of the mitochondrial and death receptors. Normal cells in the body die every day due to apoptosis, and inhibition of apoptosis results in the development of cancers, autoimmune diseases, inflammation diseases, and viral infections. B-cell lymphoma 2 (Bcl-2) family regulates apoptosis by controlling the mitochondrial pathway. The Bcl-2 family consists of anti-apoptotic proteins (such as Bcl-2, Bcl-xL, Bcl-M, and Bcl-G) and proapoptotic proteins (such as Bcl-2-associated X protein, Bcl-2 antagonist/killer, BH3-only protein, BH3 interacting domain death agonist, and Bcl-B) [49]. In some cases, the dysregulation of apoptosis occurs in cancer cells through the regulation of the transcription factor NF-κB [44,50,51]. Apoptosis-promoting activity has been proposed as an effective treatment for PCa progression.

Therefore, reducing DNA methylation in PCa is a good way to prevent the development and progression of PCa.

*Acer* species (Acereae) are a genus of trees and shrubs commonly known as maple. There are almost 128 species in the world and 17 species are native to Korea [52]. It is reported that flavonoids, tannins, phenylpropanoids and terpenoids, etc., were isolated from *Acer* species [53,54,55,56]. About 20 *Acer* species have been used in Chinese ethnomedical medicine; the expelling of wind, clearing heat, detoxifying the body, relieving rheumatism and lubricating joints, improving eyesight, treating sore eyes, reducing blood pressure, eliminating blood stasis, etc. [57]. The pharmacological activities of *Acer* species in rheumatoid arthritis, and arthralgia, eliminating stasis to resolve swelling and anti-oxidative, anti-inflammatory, anti-atopic dermatitis activity, anti-bacterial, anti-viral, anti-fungal, anti-obesity, anti-hyperglycemic and anti-tumor activities were reported [58,59,60,61]. It is reported that the *Acer pseudosieboldianum* (Pax) Komarov leaf (APL) extract shows potent anti-oxidative, anti-inflammatory, and anti-PCa proliferative activities [62,63]. This study aimed to assess the chemopreventive activities and elucidate the mechanisms of those compounds which were isolated from APL in relation to DNA methylation.

## 2. Results

### 2.1. Isolation of Phenolic Compounds along with the Novel Ellagitannin [Komaniin (14)]

Through the extraction and separation of compounds from APL, we obtained a total of **14** compounds, including a novel ellagitannin [komaniin (**14**)]. Thirteen known compounds include glucose derivatives [ethyl-β-D-glucopyranose (**3**) (Appendix A) [64] and (4R)-p-menth-1-ene-7,8-diol 7-O-β-D-glucopyranoside (**4**) (Appendix A) [65], one phenylpropanoid [junipetrioloside A (**5**) (Appendix A) [66,67], three phenolic acid derivatives [ellagic acid-4-β-D-xylopyranoside (**1**) (Appendix A) [68], 4-O-galloyl-quinic acid (**2**) (Appendix A) [69], and gallic acid (**8**) (Appendix A) [70], two flavonoids [quercetin (**11**) (Appendix A) [71] and kaempferol (**12**) (Appendix A) [72], and five hydrolysable tannins [geraniin (**6**) (Appendix A) [73,74,75], punicafolin (**7**) (Appendix A) [76,77], granatin B (**9)** (Appendix A) [76,78,79,80,81], 1,2,3,4,6-penta-galloyl-β-D-glucopyranoside (**10)** (Appendix A) [82], and mallotusinic acid (**13**) (Appendix A) [74,83,84,85,86,87] (Figure 1).

Komaniin (**14**) was obtained as an amorphous brown power. Structural data were as follows: HRFAB-MS m/z: 1217.075 [M-H${]}^{-}$ (calcd. 1217.059, $C_{55}H_{29}O_{33}$) (Appendix A) suggesting the molecular formula as $C_{55}H_{29}O_{33}$ The H 1-NMR spectrum of **14** (Appendix A) showed one glucose moiety at δ 6.65 and δ 5.62, 5.44, 5.52, 4.83, 4.83 and 4.30 and ^13^C-NMR spectrum of **14** (Appendix A) also showed six glucosyl signals at δ 90.28, 69.39, 62.51, 65.48, 71.73 and 62.88. Careful analysis of the H 1-NMR and C 13-NMR spectra, revealed one galloyl group (δ 7.16 singlet on H 1-spectrum), one hexahydroxydiphenoyl (HHDP) group (δ 6.62 and 7.08 every singlet on H 1-NMR) and one DHHDP group (δ 5.82, 6.66, 7.24 every singlet on ^1^H-NMR and 186.02 on C 13-NMR spectra), which is similar to the A form of geraniin (**6**); **14** established an ellagic acid group at δ 7.51 and 7.84 (every singlet) in the H 1-NMR spectrum and at δ 108.3, 110.62, 112.76, 136.25, 139.67, 148.13, 159.14 and 110.96, 111.65, 112.11, 132.11, 132.61, 137.75, and 159.48 in the C 13 -NMR spectrum with the characteristic upfield shifted lactone carbonyl signals at δ 159.14 and 159.48. The carbonyl of the galloyl group (δ_C_ 164.57) was correlated with glucopyranoside H-1 (δ_H_ 6.65), two carbonyls of the HHDP group (δ_C_ 165.66 and 167.94) were correlated with glucopyranoside group H-3 and H-6 (δ_H_ 5.44 and 4.83, 4.30), respectively, and two carbonyls of DHHDP group (δ_C_ 164.55 and 164.34) were correlated with glucopyranoside group H-2 and H-4 (δ_H_ 5.62 and 5.52), respectively, in the HMBC spectrum of **14** (Appendix A). Moreover, the HMBC spectrum of **14** also showed a correlation of ellagic acid C-3 (δ_C_ 132.61) with DHHDP group H-2′ (δ_H_ 5.82). The negative cotton effect (${\left[\theta \right]}_{228}$−18.46) and positive cotton effect (${\left[\theta \right]}_{262}$9.75) at a short wavelength in the CD spectrum of **14** (Appendix A) indicated that the absolute configuration of the HHDP group is R. According to the correlation of carbonyl (δ_C_ 164.34) with DHHDP group H-2′ and H-6′ (δ_H_ 5.82 and 6.66), and the correlation of carbonyl (δ_C_ 164.34) with DHHDP group H-6 (δ_H_ 7.26), suggested that the DHHDP is an (R)-configuration. In the CD spectrum, the positive cotton effect (${\left[\theta \right]}_{202}$10.29), and negative cotton effect ([$\theta {]}_{296}$−5.07, ${\left[\theta \right]}_{374}$−6.70) indicated that the absolute configuration of the DHHDP group is R. Generally, there are two forms of DHHDP group as an equilibrium mixture in an aqueous solution (A form and B form), the A form is a six-membered hemi-ketal form and the B form is a five-membered hemi-ketal form. However, only the A form of geraniin (**6**) was observed. According to the correlation of geraniin A form and ellagic acid, it is suggested that the geraniin A form was fixed and cannot convert to the B form. This ellagitannin is composed of one galloyl group at H_Glc_-1, one HHDP group at H_Glc_-3 and H_Glc_-6, one DHHDP group at H_Glc_-2 and H_Glc_-4, and H_EA_-3 and H_EA_ -4 of ellagic acid is correlated with C_DHHDP_-5 and C_DHHDP_ -3′ was tentatively named komaniin (**14**). From these results, the structure of **14** was established in Figure 2.

Komaniin (**14**)

The product was a brown amorphous powder. Structural data were as follows: HRFAB-MS, m/z: [M-H^-^], 1217.075 (calcd. 1217.059, C_55_H_29_O_33_), CD (Methanol): [*θ*]_228_: −18.46, [*θ*]_262_: 9.75, [*θ*]_202_: 10.29, [*θ*]_296_: −5.07 and [*θ*]_374_: −6.70. ^1^H-NMR (600 MHz, DMSO-d_6_ + D_2_O): δ_H_ 7.24 (1H, s, H_B_-6′’’’), 7.18 (2H, s, H_A_-2′), 7.17 (1H, s, H_A_-6′’’’), 7.17 (2H, s H_B_-2′), 7.13 (1H, s, H_B_-6′’), 7.07 (1H, s, H_A_-6′’), 6.64 (1H, s, H_A_-6′’’), 6.63 (1H, s, H_B_-6′’’), 6.54 (1H, bs, H_B_-1), 6.53 (1H, bs, H_A_-1), 6.52 (1H, s, H_A_-6′’’’’), 6.23 (1H, d, J = 1.8 Hz, H_B_-6′’’’’), 5.56 (1H, m, H_B_-3), 5.55 (1H, m, H_B_-2), 5.54 (1H, m, H_A_-2), 5.49 (1H, m, H_B_-4), 5.48 (1H, m, H_A_-3), 5.40 (1H, m, H_A_-4), 5.14 (1H, s, H_A_-2′’’’’’), 4.92 (1H, d, J = 1.8 Hz, H_B_-2′’’’’), 4.91 (1H, t, J = 10.8 Hz, H_A_-6a), 4.77 (1H, m, H_B_-5), 4.76 (1H, m, H_A_-5), 4.74 (1H, t, J = 9.6 Hz, H_B_-6a), 4.41 (1H, dd, J = 6.6, 9.6 Hz, H_B_-6b), 4.28 (1H, dd, J = 8.4, 10.8 Hz, H_A_-6b). ^13^C-NMR (150 MHz, DMSO-d_6_ + D_2_O): δ_C_ 193.7 (C_B_-5′’’’’), 191.0 (C_A_-5′’’’’), 167.6 (C_A_-7′’’), 167.5 (C_B_-7′’’), 165.4 (C_A_-7′’), 65.4 (C_B_-7′’), 164.9 (C_B_-7′’’’’), 164.8 (C_A_-7′’’’’), 164.6 (C_A_-7′’’’), 164.1 (C_B_-7′’’’), 164.0 (C_A_-7′), 163.9 (C_B_-7′), 153.8 (C_A_-1′’’’’), 148.5 (C_B_-1′’’’’), 146.9 (C_B_-3′’’’), 146.4 (C_B_-5′’’’), 145.2 (C_B_-3′, 5′), 145.2 (C_A_-3′, 5′), 145.0 (C_A_-3′’’’), 144.7 (C_A_-5′’’), 144.6 (C_B_-5′’’), 144.5 (C_B_-3′), 144.4 (C_A_-3′), 144.3 (C_A_-3′’’), 144.2 (C_B_-3′’’), 143.8 (C_A_-5′’), 143.7 (C_B_-5′’), 142.8 (C_A_-5′’’’), 139.0 (C_A_-4′), 139.0 (C_B_-4′), 138.2 (C_B_-4′’’’), 137.0 (C_A_-4′’), 137.0 (C_B_-4′’), 136.6 (C_A_-4′’’’), 135.7 (C_A_-4′’’), 135.7 (C_B_-4′’’), 128.0 (C_A_-6′’’’’), 124.9 (C_A_-1′’’), 124.7 (C_B_-1′’’), 124.3 (C_B_-6′’’’’), 124.0 (C_B_-1′’), 123.8 (C_A_-1′’), 119.5 (C_B_-1′), 119.4 (C_A_-1′), 119.3 (C_B_-1′’’’), 118.9 (C_A_-1′’’’), 116.4 (C_B_-2′’’’), 116.3 (C_B_-2′’), 116.1 (C_A_-2′’), 115.0 (C_A_-2′’’’), 114.3 (C_B_-2′’’), 114.2 (C_A_-2′’’), 112.6 (C_A_-6′’’’), 112.6 (C_B_-6′’’’), 110.2 (C_A_-2′, 6′), 109.9 (C_B_-2′, 6′), 109.7 (C_B_-6′’), 109.4 (C_A_-6′’), 108.4 (C_B_-3′’’’’), 107.3 (C_B_-6′’’), 107.1 (C_A_-6′’’), 95.3 (C_A_-4′’’’’), 91.7 (C_A_-3′’’’’), 91.6 (C_B_-4′’’’’), 91.0 (C_B_-1), 89.9 (C_A_-1), 72.5 (C_B_-5), 71.8 (C_A_-5), 69.7 (C_B_-2), 69.2 (C_A_-2), 66.1 (C_B_-4), 65.2 (C_A_-4), 63.1 (C_B_-6), 62.9 (C_A_-6), 62.5 (C_A_-3), 61.6 (C_B_-3), 51.2 (C_B_-2′’’’’), 45.5 (C_B_-2′’’’’).

### 2.2. Anti-Proliferative Activities

Compared to the other compounds, the six hydrolyzable tannins (**6**, **7**, **9**, **10**, **13**, and **14**) showed more potent anti-proliferative activities in androgen-independent prostate cancer PC-3 cells (Figure 3, Table 1). Similar to the antiproliferative activity in PC-3 cells, the six hydrolyzable tannins (**6**, **7**, **9**, **10**, **13**, and **14**) also showed potent anti-proliferative activities in androgen-dependent LNCaP cells. Moreover, all compounds, including the hydrolyzable tannins, showed stronger anti-proliferative activities in LNCaP cells than in PC-3 cells. These results suggest that the compounds isolated from APL, especially the hydrolyzable tannins, can inhibit LNCaP cells more potently than PC-3 cells (Figure 4, Table 2).

### 2.3. Apoptosis-Promoting Activities

The six hydrolyzable tannins (**6**, **7**, **9**, **10**, **13**, and **14**) showed potent apoptosis-promoting activities in PC-3 cells (Figure 5 and Figure 6). Furthermore, these hydrolyzable tannins (**6**, **7**, **9**, **10**, **13**, and **14**) showed more potent apoptosis-promoting activities in LNCaP cells than that in PC-3 cells. In addition, **14** displayed apoptosis-promoting activity later than other ellagitannins. These results suggest that the hydrolyzable tannins isolated from APL, especially ellagitannins (**6**, **9**, **13**, and **14**) in the dehydrohexahydroxydiphenic acid (DHHDP) group, inhibit LNCaP cells more potently than PC-3 cells (Figure 7 and Figure 8). 

### 2.4. Cytokine Inhibitory Activities

Inhibition of IL-6 has a role in prostate cancer growth and prostate prevention with high specificity. The six hydrolyzable tannins (**6**, **7**, **9**, **10**, **13**, and **14**) and two flavonoids (**11** and **12**) showed potent anti-oxidative and anti-inflammatory activities. These six compounds were selected for further downstream experiments. The tannins showed stronger IL-6 production inhibitory activity than that of the positive control, Bay 11-7082(BAY) and the efficiency depended on the number of galloyl groups (**10** > **7** > **6**, **9**, **13**, **14**). In addition, the (S)-DHHDP group was less potent in inhibiting IL-6 production than that of the (R)-DHHDP group, according to the results of IC50 of **9** < **6**, **13**, and **14** (Table 3).

### 2.5. NF-κB Inhibitory Activities of Hydrolysable Tannins

NF-κB inhibition was suggested as a good cancer therapy and prevention target. The four ellagitannins (**6**, **9**, **13**, and **14**) in the DHHDP group and the two hydrolyzable tannins (**7** and **10**) not in the DHHDP group were examined for their NF-κB inhibitory activities. The results showed that **6**, **9**, **13**, and **14** inhibit NF-κB activity more potently than **7** and **10**, or the positive control, BAY (Figure 9, Table 4).

### 2.6. DNA Methyltransferases Inhibitory Activities of Selected Ellagitannins

High concentrations of ellagitannins potently inhibited DNMT3a and DNMT3b protein expressions. In particular, the DHHDP group conjugated with ellagic acid group compound, **14**, showed the strongest inhibitory activity for DNMT3; **6** only inhibited DNMT1 minimally, and **13** and **14** in the (R)-DHHDP group showed more potent DNMTs inhibitory activities than that of **9** in the (S)-DHHDP group (Figure 10).

### 2.7. Methyl Removing and Re-Expression Activities of Glutathione S-Transferase P1 of Ellagitannins from APL

According to previous research, [29] clustering of CpG sites in and near the transcription start site (−216 to +7) were selected to design the primers for methyl specific (MS)-polymerase chain reaction (PCR) to detect DNA methyl removing activity. On the molecular level, the ellagitannins (**6**, **9**, **13**, and **14**) in the DHHDP group showed potent enhancement of glutathione S-transferase P1 (GSTP1) mRNA and protein expressions. In particular, the (R)-DHHDP group ellagitannins (**6**, **13**, and **14**) increased the mRNA and protein expressions of GSTP1 more than that of the (S)-DHHDP group ellagitannin (**9**) (Figure 11 and Figure 12). The mRNA and protein expressions of the unmethylated GSTP1 were enhanced, while that of the methylated GSTP1 was inhibited by the ellagitannins from APL. In particular, **13** and **14** showed more potent demethylation activity of GSTP1 (Figure 13). These results suggest that, although the DHHDP group can generally enhance GSTP1 expression, the (R)-DHHDP group, compared with the (S)-DHHDP group, might be better in PCa prevention.

## 3. Discussion

Phytochemicals are naturally occurring plant-based compounds. Dietary phytochemicals include polyphenols, carotenoids, glucosinolates, organic sulfides and terpenoids and have potential as chemical treatments. They show improved anticancer activity due to the regulation of several cell death pathways, such as apoptosis, inhibition of cell proliferation and invasion and migration [88,89]. The compounds we isolate from APL also include flavonoids and ellagitannins. Thus, compounds from APL may be good agents for prostate cancer prevention.

The compounds we isolate from APL also have components such as ellagic acid, gallic acid and quercetin. Thus, compounds from APL may be good agents for prostate cancer prevention. We isolated the **14** compounds; glucose derivatives (**3**, **4**), one phenylpropanoid (**5**), three phenolic acid derivatives (**1**, **2**, **8**), two flavonoids (**11**, **12**) and five hydrolyzable tannins (**6**, **7**, **9**, **10** and **13**) including a novel ellagitannin **14** from APL. The hydrolyzable tannins (**6**, **7**, **9**, **10, 13** and **14**) isolated from APL have potent anti-proliferative and apoptosis-promoting activity, especially the ellagitannins (**6**, **9**, **13** and **14**) in the DHHDP group. In particular, the novel compound **14**, which has a structure of ellagic acid conjugating with geraniin A form, showed the strongest chemopreventive activities among the other compounds. Ellagic acid has four hydroxyls and two lactone functional groups, which allow the removal of various ROS and reactive nitrogen species. The antioxidant properties of ellagic acid are related to anti-inflammatory action, neuroprotection, diabetes, cardiovascular disease, and protection against prostate cancer [90,91]. In our study, when comparing geraniin (**6**) with a novel compound (**14**), both compounds showed potent anti-proliferative and apoptosis-promoting activity. However, **14** showed the most potent inhibitory activity on DNA methyltransferase (DNMT1, 3a and 3b).

In this study, several hydrolyzable tannins (**6**, **7**, **9**, **10, 13** and **14**) which were isolated from APL showed potent anti-proliferative and apoptosis-promoting activity, compared with other compounds, especially the ellagitannins (**6**, **9**, **13**, and **14**). In addition, the ellagitannins(**6**, **9**, **13**, and **14**) which have a DHHDP group in their structure, showed potent IL-6 and NF-κB inhibitory activities. Among the DHHDP compounds, the novel compound **14**, showed the most potent inhibitory activity. So the following experiments were conducted mainly on ellagitannins (**6**, **9**, **13**, and **14**) in the DHHDP group. The ellagitannins (**6**, **9**, **13**, and **14**) displayed demethylation activity as measured by methyl removal, GSTP1 re-expression, and DNMT inhibitory activities. In particular, compounds **13** and **14** showed more potent activity on the demethylation activity of GSTP1 and potent inhibitory activity on DNA methyltransferase (DNMT1, 3a and 3b) and GSTP1 methyl removing and re-expression activities. In addition, DNA methylation-inducing factors, IL-6 and NF-κB, which are associated with the regulation of DNMTs, were inhibited by the ellagitannins (**6**, **9**, **13**, and **14**). According to the results of demethylation activity, it appeared that methylation of GSTP1 is associated with DNMT1. These results suggest that the ellagitannins in the DHHDP group, especially the novel compound **14**, from APL can be developed as a new agent for PCa chemoprevention. This study showed the possibility that ellagitannins have effective preventive activity against prostate cancer. Further, additional research will be tried to verify whether chemopreventive effects are also exhibited in vivo.

## 4. Materials and Methods

### 4.1. Plant Material

The leaves of the AP were collected from Seoul University Forest College of Agriculture & Life Sciences, Seoul National University in Gwangyang, Jeollanam, Republic of Korea, in September 2018. The plant was identified by doctoral researcher Park Chang-Gwoun (Seoul University Forest College of Agriculture & Life Sciences, Seoul National University). A voucher specimen (APL 2018-09) has been stored at the herbarium in the College of Pharmacy, Chung-Ang University.

### 4.2. Isolation of the Compounds from APL

The leaves of AP (1.62 kg) were extracted with 80% prethanol A (ethyl alcohol) at room temperature (25 °C) three times to obtain the extract (595.61 g). The extract was dissolved in water and filtered through Celite (Duksan Pure Chemical, Ansan, Korea), the water-soluble fraction was applied to a Sephadex LH-20 column equilibrated with water and eluted with a water-methanol gradient system and washed by 60% acetone, eleven fractions (Fr.) were yielded. Fr. 2 (10.53 g) was applied to the MCI CHP20P column with a water-methanol gradient system; Fr. 2-26 was dissolved with methanol and filtered to yield **1** (30.9 mg). Repeated column chromatography of Fr. 2-2 (1544.5 mg) on an ODS-B column with a water-methanol gradient system was conducted to yield **2** (121.9 mg). Similarly, Fr. 2-4 was subjected to an ODS semi-prep column with an MPLC system with a water-methanol gradient system to yield **3** (37 mg). Fr. 2-22 (294.9 mg) was subjected to an ODS semi-prep column with an MPLC system with a water-methanol gradient system to yield **4** (71.7 mg). Following the chromatography, Fr.2 yielded **5** (17.9 mg). Fr.7 (94.59 g) was subjected to an MCI CHP20P column with a water-methanol gradient system to yield **6** (45.53 g). Repeated chromatography of Fr. 7-2 (281.9 mg) on an ODS semi-prep column with MPLC with 25% methanol yielded **8** (24.2 mg), **9** (23.9 mg) and **13** (46.2 mg). Fr. 7-13 (2.1 g) was subjected to an ODS semi-prep column with an MPLC system and a water-methanol gradient system to yield **11** (210.1 mg). Fr. 10 (1.99 g) was applied to a Sephadex LH-20 column with a water-methanol gradient system to yield **7** (236.6 mg). Repeated chromatography of Fr. 10-7 (417 mg) on an ODS semi-prep column with an MPLC system with 30% methanol yielded **10** (49.7 mg). Fr. 7-15 (2.1 g) was subjected to an ODS semi-prep column with an MPLC system with a water-methanol gradient system to yield **12** (32.4 mg). Fr. 11 (11.30 g) was applied on MCI CHP20P and a Sephadex LH-20 column with a water-methanol gradient system to yield **14** (192.7 mg).

### 4.3. Cell Cultures

RAW 264.7 macrophage cell lines, PC-3 and LNCaP, were purchased from the Jongno-gu, Korean Cell Line Bank. Cells were incubated at 37 °C with 5% CO2 in a DMEM or RPMI 1691 medium (Corning, Corning, NY, USA) containing 10% fetal bovine serum (Welgene, Gyeongsan-si, Gyeongsangbuk-do, Republic of Korea) and 1% penicillin (Gibco, Grand Island, NY, USA) in culture flask (75 cm^2^) (SPL Life Sciences, Pocheon-si, Gyeonggi-do, Republic of Korea).

### 4.4. Cell Proliferative Inhibition Assay

The anti-proliferative activities of compounds in PC-3 and LNCaP cell lines were measured by the MTT assay. PC-3 and LNCaP cells (1 × 105 cells / well) were pre-cultured in a 96-well plate for 4 h and treated with 20 µL samples (final concentration of 100, 50, 25, 12.5 and 6.25 µM for compound level samples) and 180 µL medium for the sample group or 200 µL medium for the control group. After incubating for 48 h at 37 °C, the medium was replaced with 1 mg/mL MTT and incubated for a further 4 h at 37 °C. Then, the suspension was removed and the produced formazan was dissolved in 100 µL DMSO. After mixing gently, the optical density was measured at 540 nm using the FlexStation 3 Multi Mode Microplate Reader (San Jose, CA, USA).

### 4.5. Apoptosis-Promoting Assay

PC-3 and LNCaP cell lines (2 × $10^{6}$ cells / well) were pre-cultured for 4 h and treated with 200 µL samples (final concentrations of 100, 50 and 25 µM for compound level samples) and 1800 µL medium. The normal control group was only treated with 2000 µL medium. The experimental and control groups were then incubated for 48 h at 37 °C. The cells were collected, washed in ice-cold phosphate buffer saline (PBS), and washed once in 1 × binding buffer provided in the Apoptosis Detection kit (BD, Franklin Lakes, NJ, USA). Then, the cells were resuspended in the staining solution containing 5 µL of Annexin V-fluorescein isothiocyanate (FITC) and propidium iodide (PI) dissolved in 1 × binding buffer. The mixtures were incubated for 15 min in the dark on ice. The fluorescence was analyzed by flow cytometry (BD-LSR II, San Jose, USA) using the ‘CellQuest 2.0′ software. At least 10,000 events were recorded and presented as dot plots.

### 4.6. Cytokine Inhibition Assay

RAW 264.7 macrophage cell lines (2 × $10^{6}$ cells/well) were pre-cultured in a 6-well plate for 4 h and treated with 200 µL samples (final concentration of 100, 50 and 25 µM for compound level samples), 200 µL lipopolysaccharides (LPS, final concentration is 1 µg/mL) and 1600 µL medium. The negative control group was treated with 200 µL LPS and 1800 µL medium, and the normal control group was only treated with 2000 µL medium. After incubating for 24 h at 37 °C, the suspension of each well was moved to a 5 mL vial for storage at 4 °C.

IL-6 production was measured using a mouse IL-6 ELISA kit (Young In Frontier, Seoul, Republic of Korea). After 100 µL of samples or standard were added to the 96 well plate of the kit for 2 h at 37 °C, the suspension was removed and the wells were washed three times with 100 µL/well of washing buffer. Then they were incubated for 1 h at 37 °C after adding the diluted secondary antibody. The suspension was removed and each well was washed three times again; 100 µL of working streptavidin HRP was added. After incubating for 30 min at 37 °C, the suspension was removed and each well was washed three times. After 100 µL of the substrate was added into each well, the liquid began to turn blue. Incubating the plate at room temperature for 5 min, 100 µL stop solution was added and the absorbance was measured at 450 nm by an ELISA reader (Tecan, Salzburg, Austria). Based on the standard curve, the inhibitory effect of IL-6 production and IC50 value, which is the concentration required for 50% IL-6 production, were calculated. Bay 11-7082 (BAY) (Sigma, St. Louis, MO, USA) was used as the positive control group.

### 4.7. Western Blotting Assay

After the cells were treated with compounds (final concentration of 100, 50, and 25 µM for compound level samples), the cells were collected and solubilized in ice-cold lysis buffer (Thermo Scientific, Waltham, MA, USA) supplemented with protease (Thermo Scientific, Waltham, MA, USA). The extracted total proteins were separated by SDS-PAGE on 10 % polyacrylamide gels and electrophoretically transferred to a PVDF membrane (Bio-Rad, Hercules, CA, USA). Membranes were blocked in a blocking buffer for 1 h and subsequently incubated in a blocking buffer containing primary antibodies at room temperature for 2 h. After washing the membranes with tris-buffered saline containing 0.5 % Tween-20, the secondary antibodies were incubated for 1 h at room temperature. Then, the proteins were visualized using ECL Prime Western Blotting Detection Reagent (GE Healthcare, Chicago, IL, USA), and the band intensity was analyzed using ImageJ software (National Institutes of Health, Bethesda, MD, USA). The IC50 value, which is the concentration required for inhibition of 50% of the protein concentration, was calculated. Bay 11-7082 was used as the positive control group.

### 4.8. Real-Time Reverse Transcription Polymerase Chain Reaction

PC-3 and LNCaP cell lines (2 × 1$0^{6}$ cells/well) were pre-cultured in a 6-well plate for 4 h and treated with 200 µL of compounds from the APL (final concentration of 100, 50, 25 µM for compound level) and 1800 µL of the medium. The normal control group was only treated with 2000 µL of the medium. The experimental and control groups were then incubated for 48 h at 37 °C. Total RNA was extracted from the cells using the TRIzol reagent (Invitrogen, Waltham, MA, USA). Chloroform was added to the TRIzol reagent, and the tubes were briefly shaken. The mixtures were centrifuged at 13,500 rpm and 4 °C for 15 min. The upper phase was then transferred to a new tube containing isopropanol. After the mixtures were incubated at room temperature for 10 min, they were centrifuged at 13,500 rpm and 4 °C for 10 min. The supernatant was removed and washed with 70% ethanol. After centrifuging at 11,500 rpm and 4 °C for 10 min, the RNA pellet was briefly dried. The purified RNA was dissolved in Diethyl Pyrocarbonate (DEPC)-treated water. 1 μg of isolated RNA underwent reverse transcription with the First-Stand Synthesis System kit (Thermo Scientific, Waltham, MA, USA). After adding oligonucleotides and dNTP to the 1 μg of RNA, the mixture was incubated at 65 °C for 5 min. Following incubation, SSIV buffer, dithiothreitol, and reverse transcriptase (RT) enzyme provided in the kit were added, and the mixture was incubated at 55 °C for 10 min. The RT reaction was inactivated by incubating the solution at 80 °C for 10 min to obtain cDNA samples; 1 μL of the cDNA samples were added to 10 μL of the Universal SYBR Green Supermix (BioRAD, Hercules, CA, USA), 2 μL of 10 pM primers (1 μL each of 10 pM upstream and downstream primers), and 7 μL DEPC-treated deionized water. The real-time PCR conditions were: denaturation at 95 °C for 5 min for the first cycle, 30 s for the second cycle, 30 s of annealing at 59 °C, and 30 s of extension at 72 °C. The final extension was performed at 72 °C for 10 min. The following primer sequences were used for β-actin (187, 5′-ACCATGGATGATGATATCGC-3′; 5′-ACAGGCTGGGGTGTTGAAG-3′) and GSTP1 (5′-TCACTCAAAGCCTCCTGCCTAT-3′; 5′-CAGTGCCTTCACATAGTCATCC-3′).

### 4.9. Methyl Specific-Polymerase Chain Reaction

PC-3 and LNCaP cell lines (2 × 1$0^{6}$ cells/well) were pre-cultured in a 6-well plate for 4 h and treated with 200 µL samples (final concentration of 25, 50, and 100 µM for compound level) and 1800 µL of the medium. The normal control group was only treated with 2000 µL of the medium and then incubated for 48 h at 37 °C. The cells were collected and washed in ice-cold PBS, and the DNA was extracted using the QIAamp DNA Mini Kit (QIAGEN, Hilden, Germany). PBS, QIAGEN protease, and AL buffer from the kit were added to the collected cells. After incubating at 56 °C for 10 min, the mixture was centrifuged, and ethanol was added. Then, the mixture was centrifuged at 8000 rpm, and the collection tubes were discarded. After washing twice with AW1 buffer and AW2 buffer, AE buffer was added to elute the DNA samples. DNA bisulfite modification was performed using the MethylEasy Xceed kit (Genetic Signatures, NSW, Australia); 3M NaOH was added to the collected DNA, and the mixture was incubated for 15 min at 37 °C. After adding the combined Reagent 1 and Reagent 2 from the kit, the mixture was incubated at 80 °C for 45 min. After adding Reagent 3, the DNA was washed twice using Reagent 4. Finally, Reagent 5 was added, and the solution was incubated at 95 °C for 20 min. After DNA bisulfite modification, PCR was carried out; 2 μL of the bisulfite-modified DNA samples were added to 10 μL of the Universal SYBR Green Supermix (BioRAD, Hercules, CA, USA), 2 μL of 10 pM primers (1 μL each of 10 pM upstream and downstream primers), and 6 μL DEPC-treated deionized water. The real-time PCR conditions were denaturation at 95 °C for 5 min for the first cycle and 30 s starting from the second cycle, 30 s of annealing at 63.7 °C, and 30 s of extension at 72 °C. The final extension was performed at 72 °C for 10 min. The primer sequences for unmethylated GSTP1 were (5′-AAAGAGGGAAAGGTTTTTTTGGTTAGTTGTGTGGTG-3′; 5′-AAACTCCAACAAAAACCTCACAACCTCCA-3′) and methylated GSTP1 were (5′-GGTTTTTTTCGGTTAGTTGCGCGGCG-3′; 5′-CCAACGAAAACCTCGCGACCTCCG-3′).

### 4.10. Statistical Analysis

All data were calculated as mean ± S.D of a triplicate experiment. Data were analyzed by ANOVA (one-way analysis of variance), and the post hoc analysis was conducted by the Student–Newman–Keuls (S-N-K) test (*p* < 0.05). Data were tested by *t*-test. * indicate *p* < 0.05, ** indicate *p* < 0.01,*** indicate *p* < 0.001.

## 5. Patents

Minwon Lee, Composition for preventing, treating or improving prostate cancer comprising *Acer pseudosieboldianum* (Pax) Komarov extract or fraction thereof, KR Patent 10-2021-0151509, filed 5 June 2020.

## Figures and Tables

**Figure 1 plants-12-01047-f001:**
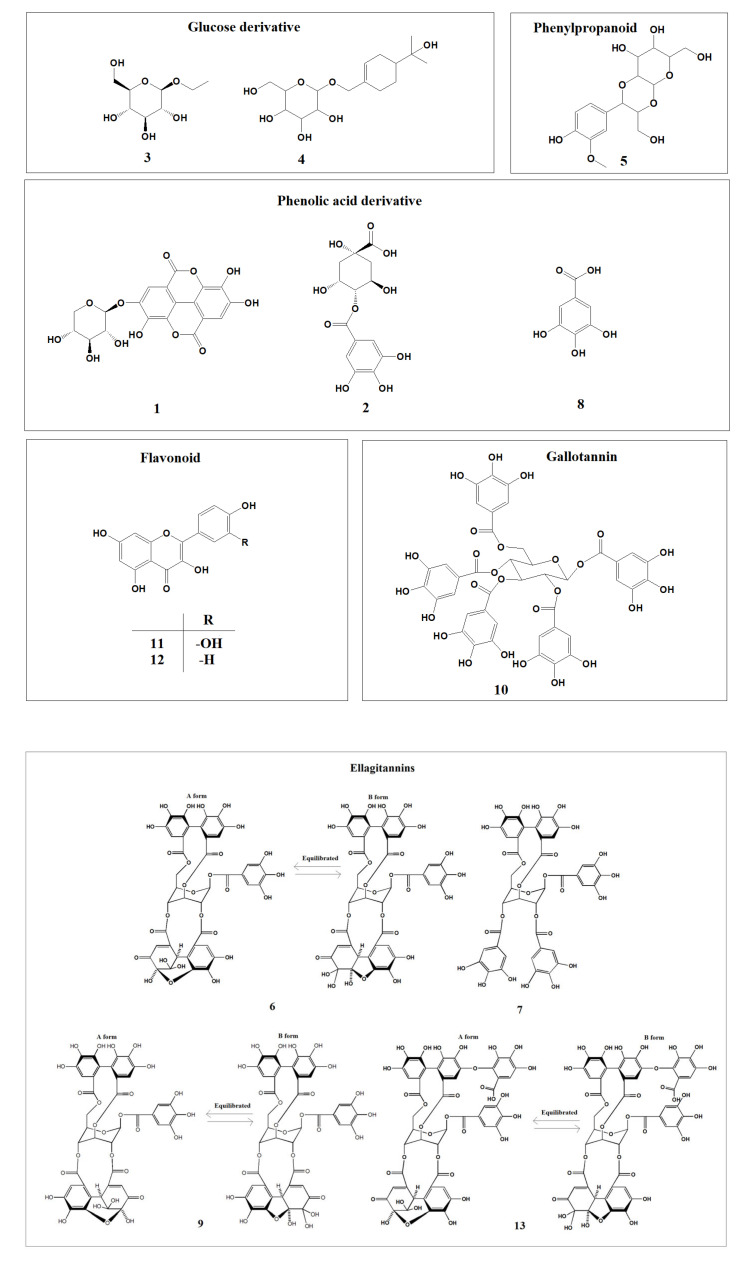
The chemical structures of compounds (**1**–**13**) isolated from APL.

**Figure 2 plants-12-01047-f002:**
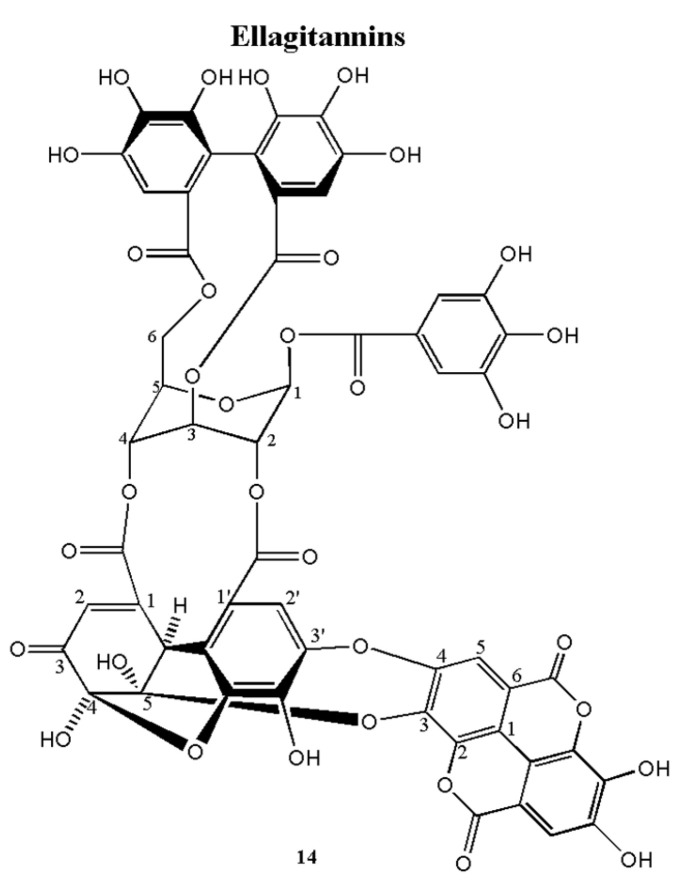
The chemical structures of **14** isolated from APL.

**Figure 3 plants-12-01047-f003:**
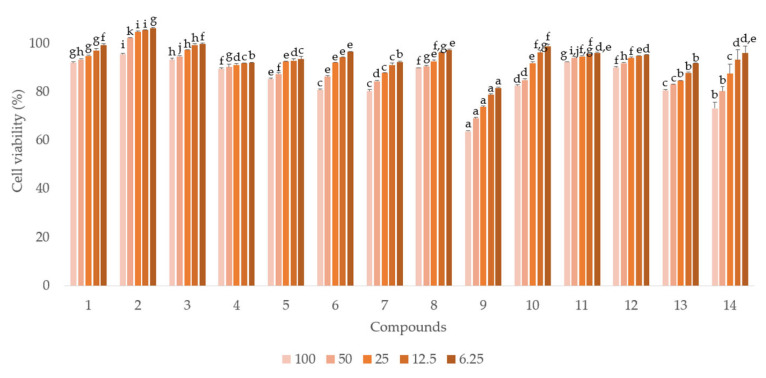
The anti-proliferative activities of compounds (**1**–**14**) isolated from APL in PC-3 cell lines. a–k indicate different superscript letters denoting significant differences (*p* < 0.05) between concentrations and groups.

**Figure 4 plants-12-01047-f004:**
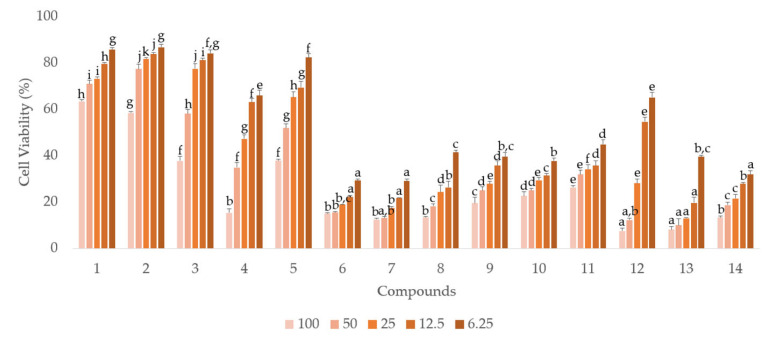
The anti-proliferative activities of compounds (**1**-**14**) from APL in LNCaP cell lines. a–k indicate different superscript letters denoting significant differences (*p* < 0.05) between concentrations and groups.

**Figure 5 plants-12-01047-f005:**
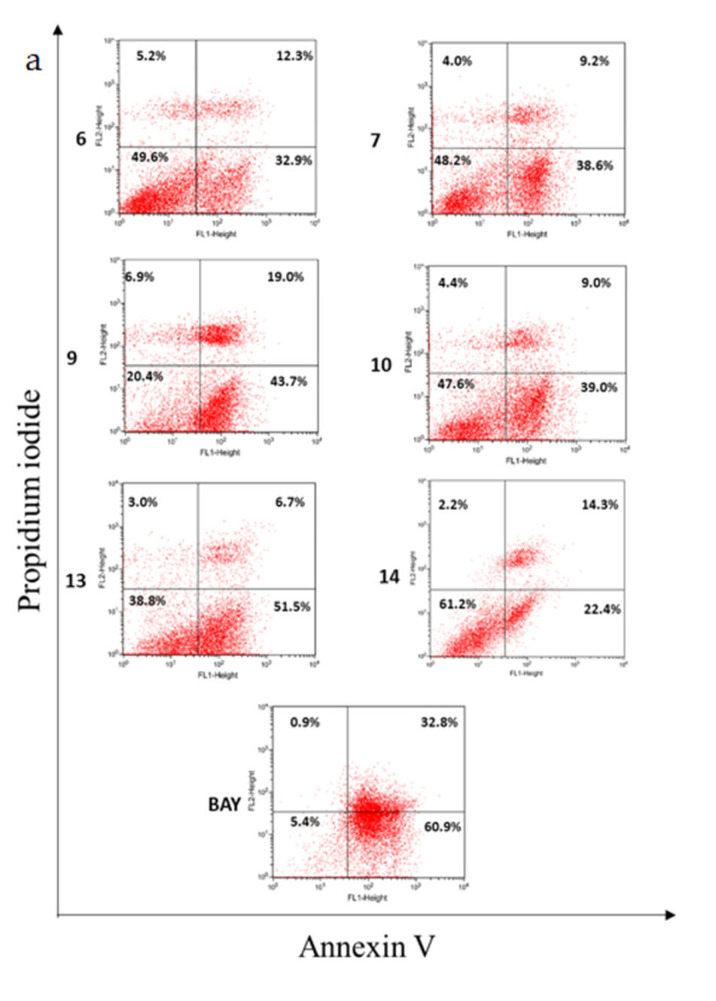
The apoptosis-promoting activities of hydrolyzable tannins (**6**, **7**, **9**, **10**, **13** and **14**) from APL in PC-3 cell lines by annexin V / propidium iodide dual staining. (**a**) 100 µM, (**b**) 50 µM, and (**c**) 25 µM of each tannin from APL were used to assess their apoptosis-promoting activities. The percentage of cells in each quadrant is indicated (upper left: necrosis; upper right: late apoptosis; lower right: early apoptosis; lower left: normal).

**Figure 6 plants-12-01047-f006:**
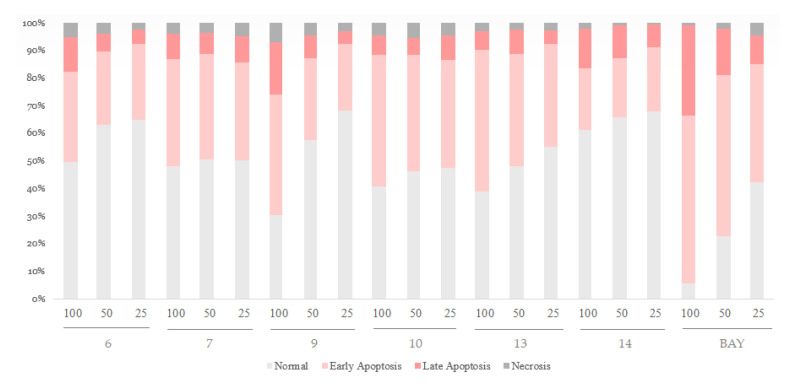
The apoptosis-promoting activities of hydrolysable tannins (**6**, **7**, **9**, **10**, **13** and **14**) from APL in PC-3 cell lines by annexin V / propidium iodide dual staining.

**Figure 7 plants-12-01047-f007:**
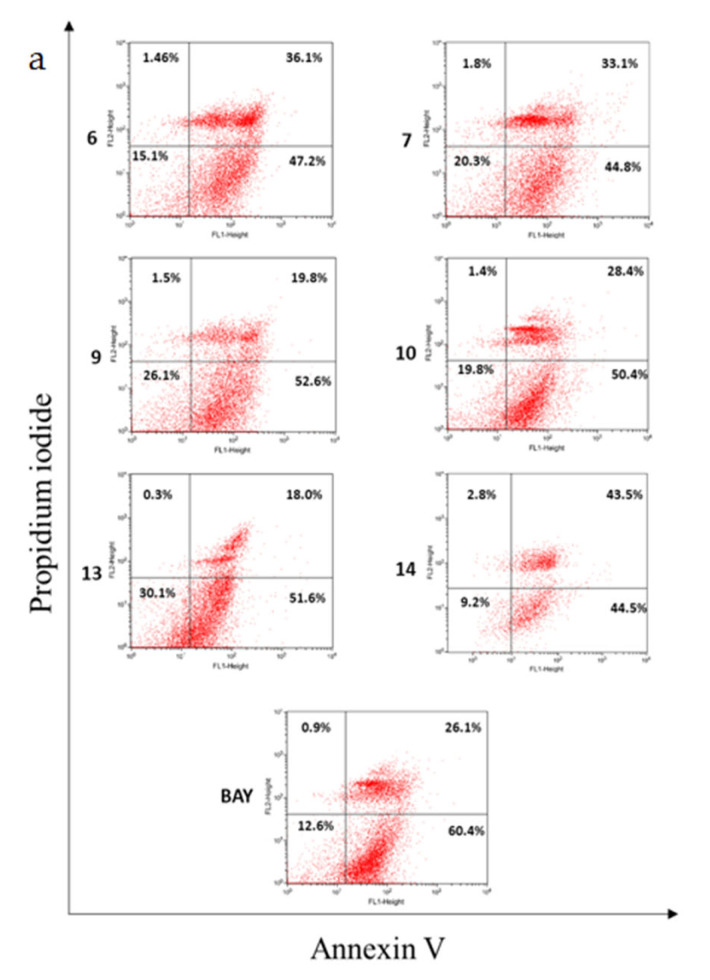
The apoptosis-promoting activities of hydrolyzable tannins (**6**, **7**, **9**, **10**, **13** and **14**) from APL in LNCaP cell lines by annexin V / propidium iodide dual staining. (**a**) 100 µM, (**b**) 50 µM, and (**c**) 25 µM of each tannin from APL were used to assess their apoptosis-promoting activities. The percentage of cells in each quadrant is indicated (upper left: necrosis; upper right: late apoptosis; lower right: early apoptosis; lower left: normal).

**Figure 8 plants-12-01047-f008:**
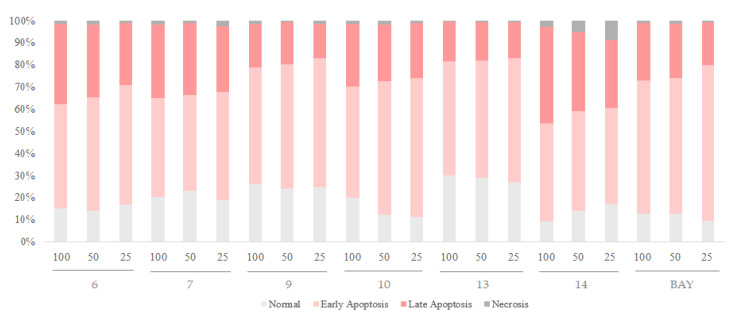
The apoptosis-promoting activities of hydrolyzable tannins (**6**, **7**, **9**, **10**, **13** and **14**) from APL in LNCaP cell lines by annexin V / propidium iodide dual staining.

**Figure 9 plants-12-01047-f009:**
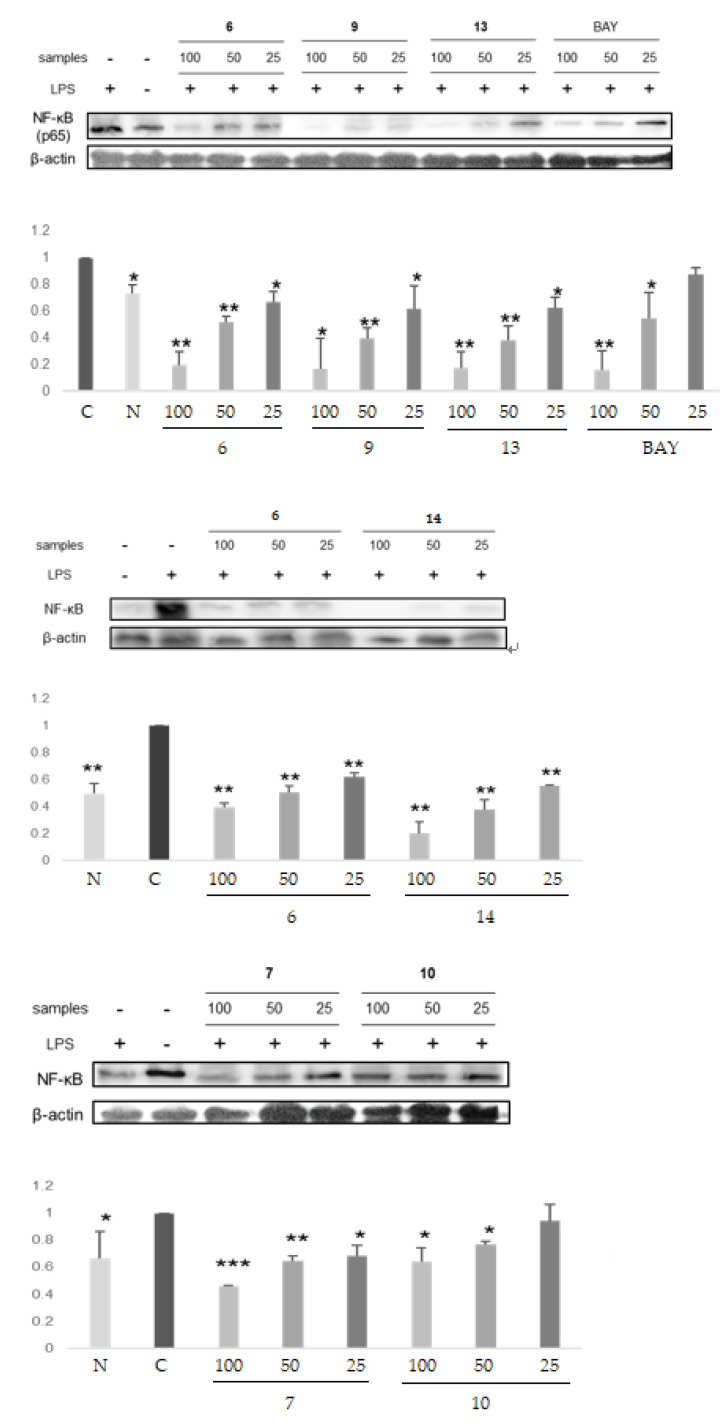
The NF-κB inhibitory activities of hydrolyzable tannins (**6**, **7**, **9**, **10**, **13** and **14**) isolated from APL. Histograms showed the densitometric data for transcription factor NF-κB normalized to the level of β-actin. The result was expressed as mean ± of triplicate experiments. * *p* < 0.05, ** *p* < 0.01, *** *p* < 0.001.

**Figure 10 plants-12-01047-f010:**
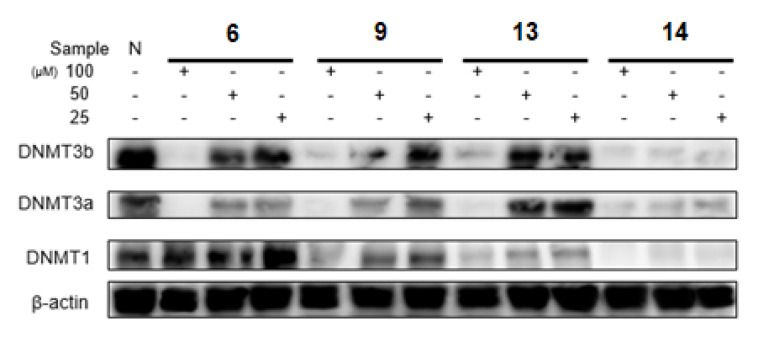
The inhibitory effect of ellagitannins (**6**, **9**, **13** and **14**) on the protein expressions of DNMTs in LNCaP cell lines.

**Figure 11 plants-12-01047-f011:**
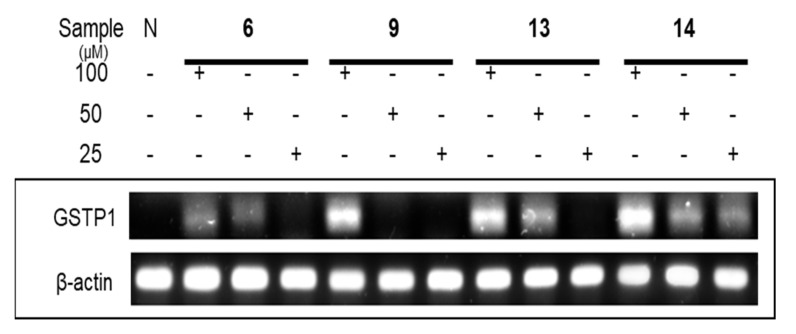
The effect of ellagitannins (**6**, **9**, **13** and **14**) on GSTP1 mRNA expressions in LNCaP cell lines.

**Figure 12 plants-12-01047-f012:**
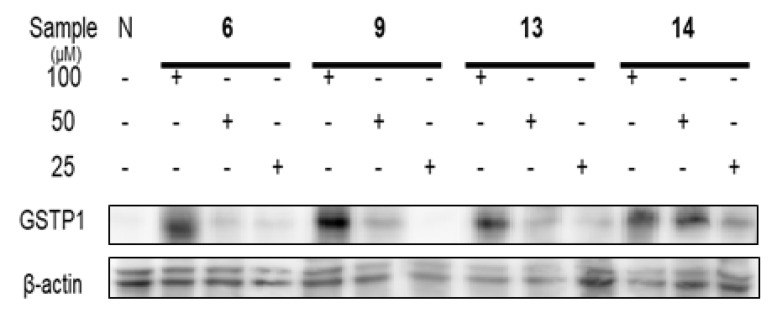
The effect of ellagitannins (**6**, **9**, **13** and **14**) on GSTP1 expressions in LNCaP cell lines.

**Figure 13 plants-12-01047-f013:**
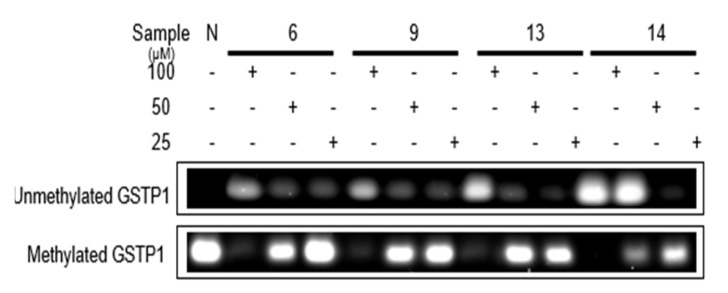
The effect of ellagitannins (**6**, **9**, **13** and **14**) on unmethylated and methylated GSTP1 expressions in LNCaP cell lines.

**Table 1 plants-12-01047-t001:** IC50 values of the anti-proliferative activities of compounds (**1**–**14**) from APL in PC-3 cell lines. The IC50 values are defined as the concentration required for 50% viability of cells. *a*–*e* indicate different superscript letters that correspond to different significant differences (*p* < 0.05).

Compounds	IC50 (µM)
**1**	>40$0^{\;e}$
**2**	>40$0^{\;e}$
**3**	>40$0^{\;e}$
**4**	>40$0^{\;e}$
**5**	>40$0^{\;e}$
**6**	271.0 ± 5.$7^{\;c}$
**7**	314.2 ± 5.$3^{\;d}$
**8**	>40$0^{\;e}$
**9**	157.1 ± 4.$1^{\;a}$
**10**	242.2 ± 10.$4^{\;b}$
**11**	>40$0^{\;e}$
**12**	>40$0^{\;e}$
**13**	308.1 ± 2.$3^{\;d}$
**14**	179.2 ± 4.$6^{\;a}$

**Table 2 plants-12-01047-t002:** IC50 values of the anti-proliferative activities of compounds (**1**–**14**) from APL in LNCaP cell lines. The IC50 values are defined as the concentration required for 50% viability of cells. *a*–*h* indicate different superscript letters that correspond to different significant differences (*p* < 0.05).

Compounds	IC50 (µM)
**1**	>10$0^{\;h}$
**2**	>10$0^{\;h}$
**3**	72.75 ± 1.9$4^{\;g}$
**4**	45.01 ± 1.2$9^{\;d}$
**5**	62.82 ± 1.3$6^{\;f}$
**6**	0.36 ± 0.0$9^{\;a}$
**7**	0.68 ± 0.1$3^{\;a}$
**8**	48.49 ± 1.2$8^{\;e}$
**9**	2.06 ± 0.4$0^{\;a}$
**10**	0.81 ± 0.1$7^{\;a}$
**11**	34.81 ± 0.7$6^{\;c}$
**12**	17.64 ± 0.6$2^{\;b}$
**13**	3.48 ± 0.4$1^{\;a}$
**14**	0.91 ± 0.1$5^{\;a}$

**Table 3 plants-12-01047-t003:** IC50 values of the interleukin (IL)-6 production inhibitory activities of the compounds (**6**, **7**, **9**, **10**, **11**, **12**, **13** and **14**) from APL. The IC50 values were defined as the concentration required for 50% inhibition of IL-6. *a*–*e* indicate different superscript letters that correspond to different significant differences (*p* < 0.05).

Compounds	IC50 (µM)
**6**	72.18 ± 7.6$7^{\;d}$
**7**	20.37 ± 4.7$7^{\;a}$
**9**	37.66 ± 5.4$3^{\;b}$
**10**	17.13 ± 1.5$1^{\;a}$
**11**	155.16 ± 10.8$3^{\;e}$
**12**	80.22 ± 2.0$8^{\;d}$
**13**	47.85 ± 2.2$9^{\;c}$
**14**	39.45 ± 4.7$6^{\;b}$
**BAY**	71.82 ± 1.8$7^{\;d}$

**Table 4 plants-12-01047-t004:** IC50 values of hydrolyzable tannins (**6**, **7**, **9**, **10**, **13** and **14**) isolated from the APL on NF-κB inhibitory activities. The IC50 values were defined as the concentration required for 50% inhibition of NF-κB. *a*–*e* indicate different superscript letters that correspond to different significant differences (*p* < 0.05).

Compounds	IC50 (µM)
**6**	52.61 ± 5.5$4^{\;b,c}$
**7**	86.90 ± 6.7$0^{\;d}$
**9**	42.37 ± 9.7$7^{\;a,b}$
**10**	119.77 ± 9.9$7^{\;e}$
**13**	42.10 ± 8.3$6^{\;a,b}$
**14**	33.86 ± 5.2$3^{\;a}$
**BAY**	61.37 ± 9.1$6^{\;c}$

## Data Availability

Data is contained within the article or Appendix A.

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
