# Peer review of "Chemopreventive Activity of Ellagitannins from Acer pseudosieboldianum (Pax) Komarov Leaves on Prostate Cancer Cells"

_plants, 2023, doi:10.3390/plants12051047_

Round 1

Reviewer 1 Report

This article isolated several compounds from Acer pseudosieboldianum (Pax) Komarov leaves and tested their anti-proliferative and apoptosis-promoting activities. Also they investigate the possible chemo-preventive mechanism of compounds. This paper is comprehensive and well written.

Below are some suggestions and questions about this article.

1.  Line 100: “Moreover, all compounds, including the hydrolysable tannins, showed stronger anti-proliferative activities in PC-3 cells than that in LNCaP cells.” According to the results, the compounds showed higher anti-proliferative activities in LNCaP cells.

2. Figure 3 & 5: a-k indicates different superscript letters that correspond to different significant differences. Using letters to indicate the statistical significance is not clear and not helpful. It didn’t provide extra information.

3. Figure 12: it looks like the loading control and GSTP1 do not come from the same assay. One of the loading control missed in the figure.

4. The compound probably cannot reach the effective concentration in vivo..

Author Response

Response to Reviewer 1 Comments

Point 1: Line 100: “Moreover, all compounds, including the hydrolysable tannins, showed stronger anti-proliferative activities in PC-3 cells than that in LNCaP cells.” According to the results, the compounds showed higher anti-proliferative activities in LNCaP cells.

Response 1: We changed the manuscript as your comment. (Line161-170)

“2.2. Anti-proliferative activities.

Compared to the other compounds, the six hydrolysable tannins (6, 7, 9, 10, 13, and 14) showed potent anti-proliferative activities in androgen-independent prostate cancer PC-3 cells. (Figure 3, Table 1) Similar to the anti-proliferative activity in PC-3 cells, the six hydrolysable tannins (6, 7, 9, 10, 13, and 14) also showed potent anti-proliferative activities in androgen-dependent LNCaP cells. Moreover, all compounds, including the hydrolysable tannins, showed stronger anti-proliferative activities in LNCaP cells than that in PC-3 cells. These results suggest that the compounds isolated from APL, especially the hydrolysable tannins, can inhibit LNCaP cells more potently than PC-3 cells. (Figure 4, Table 2)”

Point 2:  Figure 3 & 5: a-k indicates different superscript letters that correspond to different significant differences. Using letters to indicate the statistical significance is not clear and not helpful. It didn’t provide extra information.

Response 2: We showed significant differences between compound concentration and groups.

Figure 3. The anti-proliferative activities of compounds isolated from APL in PC-3 cell lines. a-k indicate different superscript letters denote significant differences (p < 0.05) between concentrations and groups.”

Figure 4. The anti-proliferative activities of compounds from APL in LNCaP cell lines. a-k indicate different superscript letters denote significant differences (p < 0.05) between concentrations and groups.”

Point 3: Figure 12: it looks like the loading control and GSTP1 do not come from the same assay. One of the loading control missed in the figure.

Response 3: We added the missing results to Compound 14 in Figure 12.

Point 4: The compound probably cannot reach the effective concentration in vivo.

Response 4: We will try in vivo experiment in further with extract level.

Reviewer 2 Report

The authors aimed assess the chemopreventive activities of APL on prostate cancer cells and elucidate the mechanisms those compounds in relation to DNA methylation.

The study covers some issues that have been overlooked in other similar topics. The structure of the manuscript appears adequate and well divided in the sections. Moreover, the study is easy to follow, but some issues should be improved. Some of the comments that would improve the overall quality of the study are:

I-) Authors must pay attention to the technical terms acronyms they used in the text

II-) Please stated the limitation of the study

Author Response

Response to Reviewer 2 Comments

Point 1 : Authors must pay attention to the technical terms acronyms they used in the text

Response 1 : We changed the terms in figure 1, 2, 5, 10, 11 and 12 to technical terms acronyms.

Point 2 : Please stated the limitation of the study

Response 2 : In the discussion section, we stated the study limitation. (Line 386-389)

“This study showed the possibility that ellagitannins have effective preventive activity against prostate cancer. Further, additional research will be tried to verify whether chemopreventive effects are also exhibited in vivo.”

Reviewer 3 Report

Comments and Suggestions for Authors

In order to be suitable for publication the paper requires revision.

INTRODUCTION

- The authors should improve the introduction.

Please emphasize the plant's critical rule and underline all the study's objectives.

There is no reference to similar studies carried out, for example, on other representatives of the Acer genus.

RESULTS

- The authors should explain the results reported in 2.2. Anti-proliferative activities in a different way.

Check the sentences “Moreover, all compounds, including the hydrolysable tannins, showed stronger anti-proliferative activities in PC-3 cells than that in LNCaP cells. These results suggest that the compounds isolated from APL, especially the hydrolysable tannins, can inhibit LNCaP cells more potently than PC-3 cells”.

The authors should move each figure near the result to improve the results reading.

- It would be necessary to improve the quality of Figure 6

- Check caption Figure 9 Where are the Histograms?

- It would be necessary to improve the quality of Figure 10

DISCUSSION

-  The authors should move some information reported in the Discussion section to the Introduction section.

-  The authors should compare the biological potential of the subject of research with other plant species. That would significantly raise the value of work.

Materials and Methods

-          The authors should improve the description of the Cell proliferative inhibition assay.

-          Authors should insert the Statistical Analysis subsection

Author Response

Response to Reviewer 3 Comments

Point 1 : Please emphasize the plant's critical rule and underline all the study's objectives. There is no reference to similar studies carried out, for example, on other representatives of the Acer genus.

Response 1 : In the introduction section, plant rule and similar study were added the genus Acer. (Line 102-111)

Acer species (Acereae) are genus of trees and shrubs commonly known as maple. There are almost 128 species and 17 species are native to Korea. [52] It is reported that flavonoids, tannins, phenylpropanoids and terpenoids etc. were isolated from Acer species. [53-56] About 20 Acer species have been used as Chinese ethnomedical medicine; The expelling wind, clearing heat, detoxifying the body, relieving rheumatic and lubricating joint, improving eye sight, treating sore eyes, reducing blood pressure, eliminating blood stasis etc. [57] The pharmacological activities of Acer species rheumatoid arthritis, arthralgia, eliminating stasis to resolve swelling and anti-oxidative, anti-inflammatory, anti-atopic dermatitis activity, anti-bacterial, anti-viral, anti-fungal, anti-obesity, anti-hyperglycemic and anti-tumor activities were reported.”

In the end of introdutcion section, we underlined the purpose of the study as below.

“This study aimed to assess the chemopreventive activities and elucidate the mechanisms of those compounds which isolated from APL to DNA methylation.”

Point 2 : The authors should explain the results reported in 2.2. Anti-proliferative activities in a different way. Check the sentences “Moreover, all compounds, including the hydrolysable tannins, showed stronger anti-proliferative activities in PC-3 cells than that in LNCaP cells. These results suggest that the compounds isolated from APL, especially the hydrolysable tannins, can inhibit LNCaP cells more potently than PC-3 cells”. The authors should move each figure near the result to improve the results reading.

Response 2: We changed manuscript as your comment. (Line 161-170)

“2.2. Anti-proliferative activities.

Compared to the other compounds, the six hydrolysable tannins (6, 7, 9, 10, 13, and 14) showed more potent anti-proliferative activities in androgen-independent prostate cancer PC-3 cells. (Figure 3, Table 1) Similar to the anti-proliferative activity in PC-3 cells, the six hydrolysable tannins (6, 7, 9, 10, 13, and 14) also showed potent anti-proliferative activities in androgen-dependent LNCaP cells. Moreover, all compounds, including the hydrolysable tannins, showed stronger anti-proliferative activities in LNCaP cells than that in PC-3 cells. These results suggest that the compounds isolated from APL, especially the hydrolysable tannins, can inhibit LNCaP cells more potently than PC-3 cells. (Figure 4, Table 2)”

We followed the rule of this journal (Figures, Tables and Schemes section)

Point 3 : It would be necessary to improve the quality of Figure 6

Response 3 : We changed the Figure 6.

Point 4 : Check caption Figure 9 Where are the Histograms?

Response 4 : We added the Histograms in the figure 9.

Point 5 : It would be necessary to improve the quality of Figure 10

Response 5: We changed the Figure 10.

Point 6 : The authors should move some information reported in the Discussion section to the Introduction section.

Response 6 : In the discussion section, chemopreventive effect which relate with DNA methylation, GSTP1, inflammation, NF-κB and Apoptosis was moved to the introduction section. (Line 45-99)

“This study aimed to evaluate the chemopreventive activities of the compounds which were isolated from APL on prostate cancer cells and elucidate the mechanisms of these compounds in relation to DNA methylation. DNMTs, including DNMT1, DNMT3a, DNMT3b, which are the family of enzymes catalyze the transfer of methyl group to DNA [15-17]. DNMT1 is the most abundant DNMTs and contribution for both de novo and maintenance methylation on tumor suppressor genes, such as GSTP1 etc. However, the active on hemi-methylated DNA of DNMT1is more potent than de novo [18-20]. The DNMT3 enzymes, including DNMT3a and DNMT3b, are mainly responsible for active on de novo methylation [21,22]. Although DNMT1 was considered to highly expressed in cancer cells, the increase in expression of de novo methyltransferases, DNMT3, also have been proved involved with cancer cells [23-25]. There are many papers proved oxidant, in-flammation, IL-6, NF-κB can increase DNMTs and inducing DNA methylation [6-14]. Thus, inhibition of these factors may the key of DNA demethylation activity.

GSTP1 belongs to Glutathione S-transferases (GSTs) enzymes family that play a key role in detoxification by conjugation of hydrophobic and electrolytic components. The GSTs are categorized into four classes, alpha, mu, pi and theta, and these enzymes effect on several factor, such as signaling pathway modulating involved in cell proliferative, progression, tumor development and recurrence. Methylation of GSTP pi gene (GSTP1) is associated with tumor development including neuroblastoma, hepatocellular carcinoma, endometrial, breast and prostate cancers. Due to researchers reported that GSTP1 are hy-permethylated in prostate cancer, while benign prostatic hyperplasia and normal prostate cells are hypomethylated [26-29]. The GSTP1 methylation of prostate cancer has been recommended as an epigenetic marker by many researchers.

Inflammation has been suggested to be a key factor in the development of several tu-mors, including PCa. It is extensively considered that the pro-inflammatory cytokines (such as tumor necrosis factor α, interleukin [IL]-6, and IL-1) and anti-inflammatory cyto-kines (such as epidermal growth factor and transforming growth factor beta) play a caus-ative role in the carcinogenesis of PCa. [30-35] IL-6 plays a key role not only in advancing tumor progression but in the early stages of carcinogenesis. Moreover, Okamoto et al. re-ported that IL-6 could enhance both androgen-dependent and independent cell (LNCaP and PC-3) growth but not the growth of benign prostatic hyperplasia [36-40]. IL-6 also in-duces DNMTs activity, thus playing a regulatory role in DNA methylation in cancer [8,10]. Therefore, the specific inhibition of IL-6 could impede PCa growth.

NF-κB is a transcription factor that mediates various functions, such as regulating DNA transcription, cytokine production, and cell survival. It also reacts with free radicals, cyto-kines, UV irradiation, and antigens [41-43]. The dysregulation of NF-κB has been as-soci-ated with inflammation, improper immune development, and cancer. Many publica-tions on different types of human tumors, including PCa, have reported that the dysregu-lation of NF-κB contributes to cell proliferation and survival. Normal cells die when nu-trients or energy are depleted through the regulation of NF-κB, but this does not occur in cancer cells owing to its anti-apoptosis activity [44-48]. Moreover, NF-κB is regulated by DNA methylation. [12] Thus, a compound that inhibits NF-κB was proposed to be an ef-fective cancer therapy and prevention agent.

Apoptosis is a form of programmed cell death, that is not similar with necrosis which cells die due to acute injury. There are two pathways in the apoptosis activation mechanism, intrinsic and extrinsic pathways, leading to the activation of the mitochon-drial and death receptors. Normal cells in the body die every day due to apoptosis, and inhibition of apoptosis results in the development of cancers, autoimmune diseases, in-flammation diseases, and viral infec-tions. B-cell lymphoma 2 (Bcl-2) family regulates apoptosis by controlling the mitochon-drial pathway. The Bcl-2 family consists of an-ti-apoptotic proteins (such as Bcl-2, Bcl-xL, Bcl-M, and Bcl-G) and proapoptotic proteins (such as Bcl-2-associated X protein, Bcl-2 an-tagonist/killer, BH3-only protein, BH3 inter-acting domain death agonist, and Bcl-B) [49]. In some cases, the dysregulation of apopto-sis occurs in cancer cells through the regulation of the transcription factor NF-κB. [44,50,51] Apoptosis-promoting activity has been proposed as an effective treatment for PCa pro-gression.”

Point 7 :  The authors should compare the biological potential of the subject of research with other plant species. That would significantly raise the value of work.

Response 7 : In the discussion section, we have added references and biological potential of other plant species as Line 282-287.

“Phytochemicals are naturally occurring plant-based compounds. Dietary phytochemicals include polyphenols, carotenoids, glucosinolates, organic sulfides and terpenoids and have potential as chemical treatments. They show improved anticancer activity due to the regulation of several cell death pathways, such as apoptosis, inhibition of cell proliferation and invasion and migration. [88,89] The compounds we isolate from APL also have such as flavonoid and ellagitannins. Thus, compounds from APL may be good agents for prostate cancer prevention.”

Point 8 : The authors should improve the description of the Cell proliferative inhibition assay.

Response 8 : We described the contents of the Cell proliferative inhibition assay (Line 455-461)

‘The anti-proliferative activities of compounds in PC-3 and LNCaP cell lines were measured by the MTT assay. PC-3 and LNCaP cells (1 x 105 cells / well) were pre-cultured in a 96-well plate for 4 hours, and treated with 20 µL samples (final concentration of 100 µg/mL, 50, 25 µg/mL, 12.5 µg/mL and 6.25 µg/mL for compound level samples) and 180 µL medium for sample group or 200 µL medium for control group. After incubating for 48 hours at 37 °C, the medium was replaced with 1 mg/mL MTT and incubated for a further 4 hours at 37 °C. Then, the suspension was removed and the produced formazan was dissolved in 100 µL DMSO. After mixing gently, the optical density was measured at 540 nm using the FlexStation 3 Multi Mode Microplate Reader (San Jose, CA, USA).’

Point 9: Authors should insert the Statistical Analysis subsection

Response 9: We added statistics Analysis subsection to the materials and method section. (Line549-552)

“ 4.10. Statistical analysis. All data were calculated as mean ± S.D of triplicate experiment. Data were analyzed by ANOVA (one-way analysis of variance), and the post hoc analysis were occurred by the Student-Newman-Keuls (S-N-K) test (p < 0.05). Data also were tested by t-test. * indicate p < 0.05, ** indicate p < 0.01,*** indicate p < 0.001. “

Round 2

Reviewer 3 Report

Please check the concentration for each experiment ( materials and methods) 

Author Response

Response to Reviewer 3 Comments

Point 1 : Please check the concentration for each experiment ( materials and methods)

Response 1 : We added the concentration for each experiment to the materials and methods. (Line 401-402 and 413-414)

“PC-3 and LNCaP cell lines (2 x  cells / well) were pre-cultured for 4 h and treated with 200 µL samples (final concentration of 100, 50 and 25 µM for compound level samples) and 1800 µL medium.”

“and treated with 200 µL samples (final concentration of 100, 50 and 25 µM for compound level samples), 200 µL lipopolysaccharides (LPS, final concentration is 1 µg/mL) and 1600 µL medium.”
